# Alcohol Abstinence Prediction on the Basis of Conservation of Resources Theory by Stevan E. Hobfoll a Study of Polish Alcohol-Dependent Persons in the Early Phase of Recovery

**Robert Modrzyński** 

Department of Clinical Psychology and Neuropsychology, Maria Curie-Skłodowska University, pl. Litewski 5, 20-080 Lublin, Poland; modrzynski.robert@gmail.com

**Abstract:** Striving for permanent alcohol abstinence can be difficult to achieve or even impossible, which in turn often results in discontinuation of treatment. The main area of interest among researchers dealing with the problem of alcohol dependence is the ability to maintain abstinence. Despite numerous studies in this area, there is still no unambiguous data on the factors affecting the recovery process of alcohol-dependent persons. The main goal of this publication is to present the Conservation of Resources Theory (COR) by S. Hobfoll as an alternative concept to understanding alcohol dependence and to answer whether maintaining abstinence can be predicted, and what kind of resources play a key role in alcohol dependence recovery. A series of two comparisons of independent variables (level, gain and loss of resources) were made in the first and sixth month after the beginning of therapy. Questionnaire longitudinal studies of 350 alcohol-dependent persons were used. Research results show that distribution of resources is of great significance in maintaining abstinence. It is important for the alcohol-dependent person's recovery process to have the opportunity to gain resources. Experiencing loss of resources in the beginning of treatment often determines their return to drinking. The project provides empirical support for research on the role of supportive factors in an alcohol-dependent person's recovery process.

**Keywords:** alcohol dependence; alcohol treatment; therapy; healing factors in therapy; resources; prediction of abstinence; conservation of resources theory

## 1. Introduction

In research on the effectiveness of dependency therapy, attention is increasingly drawn to factors affecting the recovery process of alcohol-dependent persons. When reviewing the literature, it can be noticed that there is a shortage of scientific reports on the factors that allow to predict abstinence of people undertaking alcohol dependence therapy. The main area of interest among researchers dealing with the problem of alcohol dependence is the ability to maintain abstinence [1–4].

Therapeutic programs offered to people struggling with alcohol dependence are focused on extending the periods of abstinence. In research on the effectiveness of dependency therapy, more and more often attention is being paid to variables affecting the healing process. However, when reviewing the literature, there is a shortage of scientific reports on the factors that allow to predict abstinence of people undertaking alcohol dependence therapy [5–8].

The main goal of this publication is to present the Conservation of Resources Theory by S.E. Hobfoll as an alternative concept of understanding alcohol dependence and the answer to the research question as to whether and how abstinence can be predicted on the grounds of dynamics of resources.

### 1.1. Adaptation of Therapy to the Patient's Needs

The classic therapy of alcohol dependence is currently conducted according to cognitive-behavioral and psychodynamic trends. Working with the dependent person in a spirit of motivational dialogue is also a standard.

Cognitive-behavioral therapy assumes that the most important thing is to identify dysfunctional beliefs, attitudes, and situations. Then, the next step is to work on changing mental patterns and learning new skills. Modified beliefs and skills are supposed to change the dysfunctional functioning of a dependent person [2,9].A therapist working in this field uses a number of techniques that aim at identification of beliefs, their modification and implementation (confronting and training new skills in society, outside the cabinet) [7,10]. The effectiveness of this kind of therapy indicates 33.4% of patients who, after two years from the end of the program, maintained abstinence.

A new standard in addiction therapy is the work with the patient using the motivational dialogue method. It is a client-centered, collaborative form of running a client to extract and strengthen motivation to change. It is based on four pillars—expressing empathy, developing discrepancies, following resistance, and supporting the sense of agency [1,4,11]. Working with the motivational dialogue method is not an independent form of therapy but its element. It is used to work on the patient's motivation, which seems to be a key and constant element of alcohol addiction therapy.

It would, therefore, seem that any polemic around the most effective approach to treating alcohol-dependent persons is baseless. Far more important is the complementary character of means of treatment, taking into account the answer to the question of the kind of resources a dependent person needs at particular stages of therapy.

### 1.2. Consevation of Resources Theory—Possible Contribution in Dependence Therapy

The dominating process that sets the limits of health and disease is called adaptation—a form of social and psychological accommodation to the surrounding conditions. Stevan E. Hobfoll proposes a perspective on the human being in a wider context as an individual rooted in a social structure. Stress and behaviour resulting from it are not perceived exclusively due to beliefs and resulting behavior, but also because of external and objective conditions resources. He divided these resources into four groups that make up the system. According to this system, human aspirations are carried out and significantly influence the process of deepening addiction and subsequent healing. [12,13].

S.E. Hobfoll's Conservation of Resources Theory (COR) assumes that what the human activity concentrates on(regardless of whether we are talking about an individual, family or social group) is to acquire, maintain, and protect resources—then all those objects are considered valuable. Hobfoll distinguished four types [14–17]: Material (physical objects), condition (conditioning access to other resources, e.g., health, work), personal (properties and characteristics of the person), and energy (enabling acquisition of others resources, e.g., money, time).

The general goal of all human activities is to possess and maintain material things, personality traits, living conditions, or energy. Keeping those things at a certain level decides satisfaction with life. Loss or even the threat of a loss of resources leads to the occurrence of stress. Similarly, stress may occur when the resources accumulated so far do not bring the expected profits [15,18,19]. Effective acquisition of resources leads to further effective acquisition (resource profit); analogous loss of resource leads to further losses. However, the loss of resources is incomparably more noticeable than profit. In other words, it can be said that people are dealing with the principle of "the more, the more" and "the less, the less". Hobfoll calls these phenomena 'a spiral of profits and losses of resources', and formulates the following principles:

1. Individuals with higher resources will be set up for gains in resources. Similarly, individuals with fewer resources are more likely to experience resource losses.
2. Initial resource loss will lead to resource loss in the future.
3. Initial resource gains will lead to resource gains in the future.

4.  A lack of resources will invariable lead to defensive attempts to conserve the remaining resources

Stress is one of the main factors leading to dependency and consequently hampering the effectiveness of the recovery process [1,2,4,20–22]. Stress as a reaction to difficult life events (which are often part of the history of an addict's life) is an incentive to reach for alcohol. Alcohol consumption becomes a valuable strategy over time to obtain other resources, which slowly displaces the remaining activities from the repertoire of remedial behaviors. In the absence of alcohol, getting resources becomes impossible or limited, which in turn leads to stress. For example, a shy person after drinking alcohol becomes more talkative, makes contact with others (one gains resources). This stops the process of developing remedial skills. Hence, many dependent persons feel stress due to social situations when drinking alcohol is prevented.

Empirical studies [15,23,24] clearly points out that the loss of resources is disproportionately more pronounced than profit, and thus causes more psychological stress. An addict's issues with health, professional, material problems, and disturbances in interpersonal relations grow over time. Pathological drinking leads to destructive life situations and the loss of resources, and in extreme cases, unemployment and homelessness, which blocks the possibility of compensating for resources. However, the natural need of a human is to strive for positive self-esteem, and for dependent persons, drinking is the only way to deal with it [2,6,12,15,25–27].

The main aim of this article is to examine whether, on the basis of resources dynamics, we can predict the maintenance of abstinence by alcohol-dependent persons and the relationship between the distribution of resources and their achievement of therapeutic aims. The main hypothesis assumes that maintaining abstinence by dependent persons depends on the distribution of resources.

This article is based on research project results, which highlight the significance of resource distribution for the achievement of therapeutic aims among people's dependent on alcohol at the early stages of returning to health [28].

## 2. Materials and Methods

In response to dilemmas resulting from therapeutic practice and literature, studies to investigate the relationship between the distribution of resources and maintaining abstinence were designed.

The basic tool was S.E. Hobfoll's Conservation of Resources-Evaluation (COR-E) questionnaire, translated into Polish by a team directed by Iwona Niewiadomska at the Chair of Social Psycho-prevention of JPII Catholic University of Lublin. Patients were studied twice using this questionnaire (the firstand sixthmonth of therapy).

An additional tool was the catamnestic questionnaire prepared by Robert Modrzyński containing socio-demographic data (metrics) and information on maintaining abstinence and on the course of therapy hitherto undertaken.

The analysis employed a number of statistical methods, including the *r* Pearson correlation coefficient, chi-square test, the *t*-Student test, and logistic regression. The IBM statistical package SPSS/Amos was used for this purpose.

Three hundred and fifty people were treated in 16alcohol addiction therapy centers. The research were longitudinal. In the first and sixth months since starting the therapy, the level of resources perceived, resource losses and perceived profits in resources were measured. After completing the first stage of the study, the participants expressing their consent for participation in subsequent stages left a phone number or e-mail address. The second stage (after six months) consisted of a telephone call for a katamnest questionnaire and filling out (electronically or traditionally) a questionnaire examining the distribution of resources. In total, results were obtained from 371 people participating in the first stage, 164 of whom participated in the first and second stages. The analysis included 350 results from the first stage, and 155 (90 referral questionnaires and 65 questionnaires filled by telephone) from the second. Other sets were excluded due to incomplete data filling.

## 2.1. Characteristics of the Research Group

There were 276 men and 71 women among the examined persons. The age of the subjects was between 18 and 68 years, the average age was 42 years. The most numerous group was people living in towns with up to 50,000 inhabitants (35.5%). Among the respondents, the highest percentage was people with secondary education (38.9%), followed by professional (28.6%) and higher (22.6%); the least had basic education (9.4%). A high percentage of dependent persons participating in the study were unemployed (47.7%). Slightly more than half of the respondents were married or had partners. Most of them had one or two children. There were many unmarried persons represented as well. On average, every fourth respondent did not have children. Over 2/3 of the surveyed population grew up in families where alcohol was abused (69%)—mainly by fathers, and violence was a common occurrence. The information obtained from the katamnest questionnaire indicates a slight advantage of people treated out patiently in the surveyed population. The majority of respondents in the study group were those who had previous treatment attempts, in particular those who had taken therapy for the second time. The second largest group were those who healed for the first time. The results of the katamnest questionnaire conducted after six months from the start of therapy indicated that 54.8% of the people maintained full abstinence—single contact with alcohol was indicated by 3.2%; 7.1% were sporadically drunk; 14.2% were drunk; 19.4% of patients returned to the style of drinking similar to before the therapy period.

## 3. Research

### 3.1. The Importance of Sociodemographic Resources in Maintaining Abstinence

An important variable differentiates the group of people who maintained abstinence from those who did not maintain—it is the number of previous attempts to undergo treatment and the frequency of taking medication. The number of dependent persons undergoing therapy, who later did not maintain abstinence, was significantly higher than previous treatment attempts. The average standard deviation (SD = 2.19) indicates a differentiation in the ability to maintain a subsequent abstinence among people starting therapy. People dependent on alcohol, who in their treatment history had on average undergone two or more therapy attempts, had more often declared abstinence after six months. The difference between groups is also in the amount of drugs taken (benzodiazepines) before undergoing therapy. In those who had significantly lower abstinence, there was an increase in the number of drugs taken in this group ($p = 0.019$). Significantly, abstinence was maintained by people living with their spouses ($p = 0.042$).

An analysis of logistic regression indicates the existence of two important predictors of maintaining abstinence: The number of attempts taken earlier (t1- measurement in the first month of treatment) and the number of continuing forms of therapy (t2—following the sixth month of treatment). It turns out that the greater the number of previous treatment attempts, the lower the likelihood of maintaining abstinence after six months ($\chi^2 (7) = 31.897$, p < 0.001, R2 = 0.45).

### 3.2. Predictive Function of the Level Of Resources in Maintaining Abstinence

The amount of resources at the time of the therapy does not differentiate dependent persons. Subjects who maintained abstinence and those who returned to drinking after six months showed a similar level of resources at the time when they started treatment. Statistically significant difference in the level of resources in people who abstain from abstinence and break out of it occurs after six months from the beginning of treatment. This discrepancy applies to all types of resources (Table 1).

In order to verify the significance of the level of resources in the prediction of abstinence of alcohol-dependent persons, a logistic regression model was built (Table 2). In the first step, the variables included in the score sheet were introduced to the model, which turned out to be significant for abstinence prediction, i.e., the number of attempts of treatment before the first test and the number of forms of therapy taken between the first and second measurements. In the second step, variables such

as the level of resources at the time of taking the treatment and the difference in the level of resources between the second and the first study were included. This model turned out to be statistically significant. The results obtained in the model indicate that the higher the level of resources at the time of the treatment and the greater the difference in the perceived level of resources between the second and the first test, the greater the likelihood of maintaining abstinence.

**Table 1.** Comparison of people who maintain abstinence and return to drinking due to the level of resources owned.

| Variables | Abstinency (t2) | | | | Test t | | |
| | Maintain | | Returned | | | | |
| | M | SD | M | SD | t | df | p |
| --- | --- | --- | --- | --- | --- | --- | --- |
| Level of resources (general score) | 175.703 | 39.024 | 135.00 | 38.636 | −42.31 | 84 | 0.000 |
| Material resources | 25.656 | 74.58 | 20.727 | 7808 | −26.43 | 84 | 0.010 |
| Personal resources | 55.468 | 12.035 | 44.045 | 12.080 | −38.37 | 84 | 0.000 |
| Energy resources | 37.218 | 10.377 | 29.045 | 8510 | −33.26 | 84 | 0.001 |
| Conditions | 57.359 | 15.122 | 41.181 | 16.238 | −42.48 | 84 | 0.000 |
| Differences in the level of all resources between t2 and t1 (overall result) | 21.072 | 35.456 | −60.62 | 46.187 | −25.11 | 69 | 0.014 |
| Differences in the level of personal resources | 86.00 | 12.418 | −25.00 | 15.104 | −29.95 | 69 | 0.004 |
| Differences in the level of energy resources | 40.90 | 85.46 | −1562 | 10.314 | −22.21 | 69 | 0.030 |
| Differences in the level of conditions | 61.09 | 12.519 | −1937 | 18.368 | −20.23 | 69 | 0.047 |

t1—variable measurement at the moment of starting therapy; t2—variable measurement after six months from the beginning of therapy; df—degrees of freedom.

**Table 2.** Logistic regression coefficient for the analysis of the level of resources in the prediction of abstinence of dependent persons.

| Variables | df | P | Exp(B) |
| --- | --- | --- | --- |
| **Level of resources**(overall result) | 1 | 0.030 | 1.024 |
| **Level change**(overall result) | 1 | 0.011 | 1.031 |

$\chi^2(4) = 17,063$; $p = 0.002$; R2 = 0.370; Exp(B)—exponentiation of the B coefficient.

For an in-depth analysis of the level of specific types of resources in the prediction of sustained abstinence by alcohol-dependent persons, a regression model with multiple independent variables was used. At first, significant variables concerning treatment were introduced, like the number of previous treatment attempts identified at the first study, and the number of undertaken forms of therapy in the period between the first and second study. In the second stage, we introduced the level of resources objects, personal characteristics, energies, and conditions from the first study. In the third stage, we added the difference in the level of possessed resources between the first and second study. In order to find out which of the specific resources were significant in predicting the observance of abstinence, we conducted a difference test between persons maintaining abstinence and those who did not follow the first six months of treatment. The meaning of different resource categories in the prediction of maintaining abstinence by persons dependent on alcohol was to be gained using the *U* Mann-Whitney's non-parametric test.

The logistic regression model concerning the level of material resources turns out to be statistically significant ($\chi^2$ (4) = 11.211, $p = 0.024$, R2 = 0.255); it explains about 26% of the abstinence variations of dependent persons. A significant variable apart from the number of hitherto undertaken attempts at therapy ($p = 0.035$; Exp(B) = 0.632) is the difference in level of possessed resources ($p = 0.071$; Exp(B) = 1.137). It is worth, however, pointing out that this is a relationship at the level of a tendency without statistical significance.

If there is an increase in material resources within six months of starting treatment, then the likelihood of maintaining abstinence increases. The level of these material resources, like a place to

live appropriate to the subject's needs ($p = 0.046$) and the required domestic furnishings ($p = 0.031$) significantly condition the capacity of the person undertaking treatment to later sustain abstinence.

The level and growth of personal resources are statistically significant ($\chi^2$ (4) = 18.079, $p = 0.001$, R2 = 0.389). Based on the results, it can be argued that the higher the level of personal resources at the beginning of therapy, the more the probability of maintaining abstinence increases ($p = 0.005$; Exp(B) = 1.102).This concerns the sense that one is doing well ($p = 0.012$), hope ($p = 0.42$), the sense that one's future success is in one's hands ($p = 0.030$), not succumbing to a routine ($p = 0.022$), organisational skills ($p = 0.029$), sense of commitment ($p = 0.048$), sense of one's own value ($p = 0.041$), and awareness of the purpose to which one moves in life ($p = 0.048$).

The results of further analyses indicate that the higher the level of conditions available to alcohol-dependent persons at the start of therapy, the more the probability of maintaining abstinence ($\chi^2$ (3) = 9.197, $p = 0.027$, R2 = 0.213) increases. The possibility of increasing energy resources during the course of treatment has a positive effect on maintaining abstinence ($p = 0.031$). The higher the increase in the level of energy resources, the greater the probability that a given person will maintain abstinence. The significant predictors help with childcare ($p = 0.049$), appropriate work status ($p = 0.011$) and stable employment ($p = 0.019$).

Increasing the level of energy resources during treatment positively influences maintaining abstinence ($p = 0.031$). The higher the increase of energy resource level, the higher the probability that the subject will maintain abstinence. The level of significance was reached by resources like strength to realise undertaken projects ($p < 0.001$) and health insurance ($p = 0.027$).

### 3.3. Predictive Functions of Perceived Resource Gain in Maintaining Abstinence in People Addicted to Alcohol

The accepted testing strategy for predictive functions of perceived resource gains is analogous to the analysis concerning the importance of the level of owned resources in the prediction of abstinence in alcohol-dependent persons.

In the first stage, people maintaining abstinence and those returning to drinking in the aspect of perceived gain in resources were compared to each other. The obtained results clearly indicate a significant difference between the study groups in the 6th month after starting the therapy (Table 3). The differences relate to the overall result and the gain in personal resources and conditions. Persons who maintained abstinence declared a significantly larger increase in resources than those who returned to drinking.

**Table 3.** Comparison of people who maintain abstinence and return to drink in terms of a sense of profit in their resources.

| Variables | Abstinency t2 | | | | Test t | | |
| --- | --- | --- | --- | --- | --- | --- | --- |
| | Maintain | | Returned | | | | |
| | M | SD | M | SD | t | df | p |
| Gain in resources (overall result) | 157.74 | 61.22 | 128.14 | 49.43 | −2.01 | 85 | 0.047 |
| Personal resources | 52.87 | 18.15 | 43.71 | 15.30 | −2.08 | 85 | 0.040 |
| Conditions | 52.22 | 21.03 | 40.23 | 18.20 | −2.34 | 85 | 0.021 |
| Difference in perceived resource gains between t1 and t2 (overall result) | 19.90 | 56.13 | −10.80 | 65.36 | −2.08 | 83 | 0.040 |
| Increase in the gain in personal resources | 9.35 | 19.25 | −1.66 | 20.46 | −2.24 | 83 | 0.028 |
| Increase in the gain in conditions | 6.34 | 18.94 | −5.04 | 21.59 | −2.30 | 83 | 0.023 |

The next stage of the study covered analysis of the possibility of predicting ability to maintain abstinence on the basis of the results concerning perceived gains. A logistic regression model was created for this purpose to which were introduced the following variables: number of previous treatment attempts, number of undertaken forms of therapy in the period between the first and second measurement, perceived resource gains during the first study and the difference in perceived gains between the second and first measurement.

A vital role in sustaining abstinence in the first months following the start of therapy is played by the gain in personal characteristic resources ($p$ = 0.012; Exp(B) = 1.052). It is key that a person in therapy senses a growing improvement in the area of personal characteristic resources during its course. The logistic regression model that considers the perceived increase in personal characteristic resources between the first and second measurement has been shown to be a significant predictor of abstinence in the sixth month following the start of therapy ($p$ = 0.012). The greater the perceived increase in personal characteristic resources during the course of treatment, the higher the chance of the dependent person maintaining abstinence. Important personal characteristic resources include the sense of being valuable to others ($p$ = 0.019), a sense of realizing one's own aims ($p$ = 0.031), hope ($p$ = 0.007), sense of optimism ($p$ = 0.032), and a sense of having one's life under control ($p$ = 0.001).

An important predictor of abstinence among the examined persons is the possibility of acquiring conditions during the treatment period ($p$ = 0.205, Exp(B) = 1.022). If there is an increase in these resources during the first six months of therapy, then the chance of maintaining abstinence is increased. The condition resources playing a significant part at the start of therapy include the health of one's own children ($p$ = 0.019), health of spouse or partner ($p$ = 0.019) and help with childcare ($p$ = 0.023). During therapy, what is supportive for abstinence is increasing perceived gain in good relations with children ($p$ = 0.037), appropriate work status ($p$ = 0.004), being part of an organisation that allows one to share one's interests with others ($p$ = 0.009), stable employment ($p$ = 0.005) and recognition of one's achievements ($p$ = 0.043).

## 3.4. Predictive Functions of The Sense of Resource Loss in Maintaining Abstinence in Alcohol Dependent Persons

A similar comparative strategy for the abstinence maintaining group and the relapses revealed a significant difference in the scope of perceived loss of resources (Table 4). Persons who broke abstinence within the period of six months from the beginning of therapy more often significantly experienced a general loss of resources (M = 80.6, SD = 59.3), in particular, personal resources (M = 28.9, SD = 20.2). Maintaining abstinence in the first months of recovery is also associated with experiencing losses; however, their overall level is much less pronounced (M = 43.6, SD = 57.3)

**Table 4.** Comparison of people with abstinence and those returning to drinking in the sense of loss of resources.

| Variables | Abstinency t2 | | | | Test t | | |
| | Maintain | | Returned | | | | |
| | M | SD | M | SD | t | df | p |
| --- | --- | --- | --- | --- | --- | --- | --- |
| Sense of resource loss (overall result) | 43.68 | 57.35 | 80.61 | 59.31 | 2.539 | 83 | 0.013 |
| Perceived loss of personal characteristics | 12.31 | 18.37 | 28.90 | 20,23 | 3.502 | 83 | 0.001 |
| Reduction in the sense of the loss of personal resources between t1 and t2 | −28.80 | 26.33 | −12.47 | 21.11 | 2.467 | 79 | 0.016 |

The results of the logistic regression model confirm the possibility of predicting abstinence based on experiencing losses before undergoing the therapy. At the moment of starting treatment, the spiral of previously experienced losses determines which of the patients will return to the addiction. Based on the obtained results, it can be stated that the smaller the sense of experienced resource losses at the beginning of treatment, the greater the chance of maintaining abstinence. The model indicates that an important predictive variable is the change in severity of experienced losses. If, during the therapy, the feeling of experiencing the loss of resources decreases, the likelihood of maintaining abstinence increases (Table 5).

The perception of loss of material resources during the first test is relevant to the ability to predict abstinence. The model taking into account both the loss of resources and change in the severity of losses during therapy clearly indicates the role in maintaining abstinence, which is the loss of material

resources ($\chi^2$ (4) = 10.895, *p* = 0.028, R2 = 0.232). The subjective sense of material resource loss, in particular those like work tools (*p* = 0.001) and owning more clothes than one needs (*p* = 0.60), determines later maintenance of abstinence.

**Table 5.** Logistic regression coefficient for the sense of loss of resources in the prediction of abstinence in alcohol addicts.

| Variables | df | p | Exp(B) |
|---|---|---|---|
| The number of previous treatment attempts | 1 | 0.017 | 0.584 |
| The feeling of losing resources (general result) | 1 | 0.026 | 0.984 |
| Reduction in the spiral of loss of resources t1-t2 (overall result) | 1 | 0.005 | 0.982 |

$\chi^2(4)$ = 14.334; *p* = 0.006; R2 = 0.294.

In the case of perceived loss of personal resources ($\chi^2$ (4) = 18.620, *p* = 0.001, R2 = 0.370) and perceived loss of energy resources ($\chi^2$ (4) = 11.505, *p* = 0.021, R2 = 0.241), we observe similar relationships as in the case of experiencing losses in material resources. Both the sense of loss of personal resources and the felt energy losses have a decisive influence on the effects of therapy in a patient undertaking treatment. Persons who declare reduced intensity of energy losses have a greater chance of maintaining abstinence as regards time at work (*p* = 0.016), health insurance (*p* = 0.044) and personal characteristics resources, like a sense of one's worth to others (*p* = 0.004), a sense of realising one's purpose (*p* = 0.031), hope (*p* = 0.013), not falling into a routine (*p* = 0.010), organisational skills (*p* = 0.033), positive feelings as regards oneself (*p* = 0.009), a sense that one is doing well (*p* = 0.027) and a sense of pride in oneself (*p* = 0.051).

The ability to maintain abstinence during the first six months from the start of therapy is affected by the decreasing level of conditions ($\chi^2$ (4) = 11.394, *p* = 0.022, R2 = 0.239). This applies in particular to resources such as recognition for achievements (*p* = 0.007) and vitality/strength (*p* = 0.011).

## 4. Discussion

The conducted analyses provide the basis for applying the assumptions of Conservation of Resources Theory in the therapy of alcohol dependency. These studies draw attention to the role of environmental factors, so often neglected but supporting the recovery process of addicts.

The study results have allowed us to pinpoint the factors playing a predictive role in the capacity to maintain abstinence.

1.  The lower the level of personal characteristic, condition, object and energy resources declared by a subject at the time of undertaking therapy, the greater the probability of relapse to the earlier drinking style.
2.  Object resources are a significant determining factor in relapse. The experience of an improvement or sudden deterioration in material situation at the moment of undertaking treatment indicates a large probability of failing to maintain abstinence in the future.
3.  The possibility of an increase of resources in the course of therapy supports the maintaining of abstinence. What is key is a rise in the level of available personal characteristic resources, like a sense of success, hope, influence on the future, not falling into a routine, organizational skills, a sense of involvement, a sense of one's own value, an awareness of purpose in life, and awareness that life makes sense—a sense of independence, peace and a sense of realizing one's own goals.
4.  The sense of increase in resources is a positive factor influencing the maintaining of abstinence by alcohol-dependent persons. Of specific significance in this context is a noticeable improvement in personal characteristics resources (a sense of being valued, a sense of realizing one's own purposes, hope, optimism, a sense of having control over one's own life, self-discipline, self-pride and a sense of one's own value) and conditions resources (good relations with children, a stable

job, appropriate professional status, support in realizing work projects, understanding on the part of the employer, recognition of achievement, company and membership of an organization).

5. The experience of resource loss during the course of therapy has a decisive influence as regards maintaining abstinence.

The gained results on the one hand confirm the relevance of Hobfoll's COR theory, while on the other empirically support current analysis on the basis of Polish studies and enrich them with specific tools for therapeutic work [29,30]. The analyses above at the same time constitute an empirical support for research into the role of factors that promote the process of alcohol-dependent persons' return to health [2–5,12,31].

According to the COR, one's motivation to act is the result of a favorable distribution of resources. People are not only striving to obtain resources that ensure their survival, but are also focused on the desire to improve their own situation, which ensures their optimal functioning in the physical, material and spiritual realms. Man's efforts are directed mainly towards retaining adaptation resources and, secondly, on winning awards. The basic condition for triggering behavior is having a minimum pool of resources. Similarly, human behavior is recognized in the context of W.R. Miller and S.Rollnick's motivation theory, who emphasize that the necessary condition for change is having the possibility of introducing it. Beginning therapy by a dependent person is a decision to change. To be considered as a viable option, a certain basic level of resources is necessary. In order for the change to be effective, it is necessary to acquire further, important resources for the person. The research confirms the assumption that the ability to maintain abstinence depends on the possibility of acquiring resources. This is an important contribution to the development of studies on predicting abstinence, because specific resources have been identified to condition abstinence on the part of patients during therapy. The message of the obtained results becomes the necessity to change the therapeutic approach, from the level of psychological interactions to the level of systemic cooperation of various organizations. Therapeutic care after the program should focus on enabling the person to acquire new resources that improve the adaptation process.

The COR theory and studies confirming it [15,18,32–34] clearly indicate that the resources that people have at their disposal have basic significance for the way it functions. According to the COR, the source of motivation is a successful course of gain and loss. People are motivated to preserve the amount of resources that will ensure not only survival, but also optimal functioning in the physical, social and spiritual realms. An important part of human activities is the desire to improve their own situation, but the key condition is to have a specific pool of resources that will enable the launch of a spiral of gain. In the absence of a basic pool, a person accepts a defensive attitude, aimed solely at protecting what he already has, without any motivation to change. This seems to coincide with the motivation theory of Miller and Rolnick, who emphasize that the importance of seeking change and making efforts to implement it is possible when we see the way and the opportunity to do it [35–37]. What, according to Hobfoll, represents this opportunity is the level of resources and effective acquisition.

The decision to undertake therapy by alcohol-dependent persons is in fact a decision to change. To be considered as a real possibility at all, a base level of resources is necessary. For the change to be effective, it is necessary to increase the level of resources, because, as research has shown, the ability to maintain abstinence depends on the ability to acquire them. Therefore, the obtained results are a kind of novelty in the study of the possibility of predicting abstinence [29,31,38–40]. Among the many resources, it was possible to identify those that are particularly important for predicting the maintenance of abstinence of alcohol addicts taking up therapy.

Taking into account the variable in the form of the level of resources possessed is a pioneering contribution to the development of the COR [14–17,41,42]. Analysis of the relationship between two variables—the level of resources possessed and the sense of their gaining—allows to separate them into two separate factors, taking a significant share in the process of distributing resources. Resource gains

affect more the sense of change in resources, while the level of resources and their growth indicates their real condition.

Analysis of the results accentuates the fact of the existence of a group of alcohol-dependent persons who do not take advantage of therapy based on total abstinence and will not achieve a therapeutic aim formulated as such [8,30,40,43]. Taking into account the cost and indicator of this type of treatment's effectiveness, it becomes justified to seek criteria supporting the classification of patients to a standard program of treatment or to an alternative form of help that is the harm-reduction approach. The thesis is also in line with current postulates contributed by authors advocating the harm-reduction approach [38,44–46].

The results of the study show that it is possible to work more effectively with alcohol-dependent persons. The area of work should not, however, be limited to therapeutic interactions. To improve the quality of life of dependent persons, they not only need to change the perception of the world and of themselves and re-evaluate reality, but also feel a sense of stability in all types of resources. Over time, the patient begins to record profits in most areas of life, which significantly increases the chance of maintaining a healthy lifestyle. This is an indication for therapists and the care system—to take care for the dependent persons in as many aspects of life as possible.

The limitation of the research project was the small number of subjects who took part in two phases and that most of the respondents were men. There were difficulties during the study concerning limited access to the dependent persons following six months from the moment of treatment initiation.

In further studies, there ought to be a change in qualification procedure for participants in the project i.e., by gaining the subjects' agreement to contact them later at their place of residence. The involvement of a greater number of persons would help achieve more effective information gathering from the respondents.

**Author Contributions:** R.M. conceived and designed the studies, analysed the data and wrote the paper.

**Funding:** This research received no external funding.

**Conflicts of Interest:** The authors declare no conflict of interest.

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
