# Peer review of "Alcohol Abstinence Prediction on the Basis of Conservation of Resources Theory by Stevan E. Hobfoll a Study of Polish Alcohol-Dependent Persons in the Early Phase of Recovery"

_psych, doi:10.3390/psych1010023_

Round 1

Reviewer 1 Report

The manuscript has many merits and puts a new light on the factors determining the efficacy of the process of maintaining abstinence among Polish alcohol dependent persons. Author presents an interesting concept of explaining the maintenance of alcohol abstinence based on the Conservation of Resources Theory by Stevan Hobfoll.  Alcohol is a way of gaining different resources and solving everyday problems and resources, indeed, are the main issue of the presented manuscript. In the theoretical part, Author discusses hindrances which alcohol dependent persons have to face as well as factors which are helpful for their recovery. This part  confirms a broad knowledge of the Author. I would  suggest adding more contemporary literature (>2015) on this topic. The Conservation of Resources Theory is explained in a detailed way. Author highlights its importance in prediction of maintenance of alcohol abstinence as well as in adaptation processes in general population. 

A large group of people have been examined (N=350) in a longitudinal project consisting of 2 steps. The results are presented clearly. 

Most of examined persons were men and it might have had consequences for obtained results. I would reccomend adding more information concering this issue in "Discussion" section. 

In the "Materials and method" section the methods are not described sufficiently.

Gaining and loosing of resources relates, in my opinion,to the developmental aspect of profiting from chances and experiencing looses. Author examined persons representing all phases of adulthood. Balance between developmental chances and looses fluctuate in the course of time. In childhood, chances prevail over looses, in early adulthood chances still dominate but looses start to appear more frequently. In late adulthood looses are more frequent than chances, although there are new experiences which can be regarded as developmental chances. New social roles and demands which people are to fulfill in different phases of their lives might be closely connected with the topic of distribution of resources. Due to this, I would suggest to add a commentary about potential different psychological situation and scope of experiences of people who were examined.

Other remarks:

Although I am not a native English- speaking, I suggest minor language/stylistic amendments: alcohol instead of “alkohol” (in abstract) its element – instead of “it’s element” (p.2). Page 3: "In these studies, it was assumed that human adaptation is understood as the manner and quality of adaptation and that it depends on the successful distribution of resources".

Page 4: "In response to dilemmas resulting from therapeutic practice and literature, studies to investigate the relationship between the distribution of resources and maintaining abstinencewere designed" 

should be:

In response to dilemmas resulting from therapeutic practice and literature, studies to investigate the relationship between the distribution of resources and maintaining abstinence were designed.

In tables: Test t instead of T

Author Response

Reviewer 1

Point 1: In the theoretical part, Author discusses hindrances which alcohol dependent persons have to face as well as factors which are helpful for their recovery. This part  confirms a broad knowledge of the Author. I would  suggest adding more contemporary literature (>2015) on this topic.

Response 1: The literature has been supplemented with contemporary research

Point 2: Most of examined persons were men and it might have had consequences for obtained results. I would reccomend adding more information concering this issue in "Discussion" section. 

Response 2: Paragraph on the limitation of research has been added in which information on this subject is included

Point 3: In the "Materials and method" section the methods are not described sufficiently.

Response 3: The attention of the reviewer was taken into account and the methods used were described in detail

Point 4: Gaining and loosing of resources relates, in my opinion,to the developmental aspect of profiting from chances and experiencing looses. Author examined persons representing all phases of adulthood. Balance between developmental chances and looses fluctuate in the course of time. In childhood, chances prevail over looses, in early adulthood chances still dominate but looses start to appear more frequently. In late adulthood looses are more frequent than chances, although there are new experiences which can be regarded as developmental chances. New social roles and demands which people are to fulfill in different phases of their lives might be closely connected with the topic of distribution of resources. Due to this, I would suggest to add a commentary about potential different psychological situation and scope of experiences of people who were examined.

Response 4 Due to the breadth of the comparative material and the main goal, author limited publication to data that presents dynamics of resources In abstinence prediction. Rewiever suggestion requires taking into account in the research additional variables/methods concerning human development phases.

Point 5: Although I am not a native English- speaking, I suggest minor language/stylistic amendments: alcohol instead of “alkohol” (in abstract) its element – instead of “it’s element” (p.2). Page 3: "In these studies, it was assumed that human adaptation is understood as the manner and quality of adaptation and that it depends on the successful distribution of resources".

Page 4: "In response to dilemmas resulting from therapeutic practice and literature, studies to investigate the relationship between the distribution of resources and maintaining abstinencewere designed" 

should be:

In response to dilemmas resulting from therapeutic practice and literature, studies to investigate the relationship between the distribution of resources and maintaining abstinence were designed.

In tables: Test t instead of T

Response 5 Language/stylistic  errors have been corrected

Reviewer 2 Report

Presented for a  review text: "Theory by Stevan E. Hobfoll. A study of Polish alcohol dependent persons in the early phase of recovery"  is a very interesting and valuable text. It contains innovative research that refers to the theory of the Distribution of Resources by S. Hobfoll to the practice of addiction treatment in the context of acquistion of factors affecting the recovery process of people addicted to alcohol. This work perfectly complements the existing shortcomings in the literature  in the field of alternative forms of therapeutic programs. The text presents a high level of substantive, methodological and statistical preparation of the author to conduct and present scientific research. Careful preparation of the text should be however supported before publishing by introducing several modifications:

 A research question, which in its present form is not correctly formulated, needs to be corrected: "(...) research question: on the basis of the dynamics of resources, whether and how can abstinence be predicted"
It is not correct to formulate a research question as being both closed (whether?) and open (how?). It is worth refining this aspect of the text as well as separating in the methodological part of the work a separate place for the research question. In the current form, the formulation of the question only in the abstract part of a manuscript is insufficient.

The author presents a rich review of the subject literature but does not formulate a research hypothesis based on it. The research hypothesis is an important element of the research work and its formulation should not be avoided. I suggest that Author considers the introduction of a research hypothesis.

 The age of the participants was included in the range 18-68 years, indeed the examined persons represented all three developmental periods of adulthood: early, medium and late. This important development aspect is worth emphasizing in the description part of the study group. In addition, in theoretical part it is worth referring to development tasks as a resource for effective planning of work with a patient addicted to alcohol.

 The discussion of the results seems to be poor in reference to the cited literature, it needs to be supplemented

References should be enriched with additional English-language positions that are relevant to the subject of work

The text is undoubtedly worth publishing, which I recommend.

Author Response

Reviewer 2

Point 1:   A research question, which in its present form is not correctly formulated, needs to be corrected: "(...) research question: on the basis of the dynamics of resources, whether and how can abstinence be predicted"
It is not correct to formulate a research question as being both closed (whether?) and open (how?). It is worth refining this aspect of the text as well as separating in the methodological part of the work a separate place for the research question. In the current form, the formulation of the question only in the abstract part of a manuscript is insufficient.

The author presents a rich review of the subject literature but does not formulate a research hypothesis based on it. The research hypothesis is an important element of the research work and its formulation should not be avoided. I suggest that Author considers the introduction of a research hypothesis.

Response 1: The author reformulated the research question. It was also detailed along with the hypothesis in the final theoretical part.  

Point 2:  The age of the participants was included in the range 18-68 years, indeed the examined persons represented all three developmental periods of adulthood: early, medium and late. This important development aspect is worth emphasizing in the description part of the study group. In addition, in theoretical part it is worth referring to development tasks as a resource for effective planning of work with a patient addicted to alcohol. 

Response 2:  Due to the breadth of the comparative material and the main goal, author limited publication to data that presents dynamics of resources In abstinence prediction. Rewiever suggestion requires taking into account in the research additional variables/methods concerning human development phases.

Point 3:  The discussion of the results seems to be poor in reference to the cited literature, it needs to be supplemented

Response 3:  The discussion of results was supplemented according to the recommendations of the reviewer

Point 4: References should be enriched with additional English-language positions that are relevant to the subject of work

Response 4: The literature has been supplemented with contemporary research